# Upstream modes and antidots poison graphene quantum Hall effect

N. Moreau[1], B. Brun [1], S. Somanchi[2], K. Watanabe [3], T. Taniguchi [4], C. Stampfer [2] & B. Hackens [1✉]

The quantum Hall effect is the seminal example of topological protection, as charge carriers are transmitted through one-dimensional edge channels where backscattering is prohibited. Graphene has made its marks as an exceptional platform to reveal new facets of this remarkable property. However, in conventional Hall bar geometries, topological protection of graphene edge channels is found regrettably less robust than in high mobility semiconductors. Here, we explore graphene quantum Hall regime at the local scale, using a scanning gate microscope. We reveal the detrimental influence of antidots along the graphene edges, mediating backscattering towards upstream edge channels, hence triggering topological breakdown. Combined with simulations, our experimental results provide further insights into graphene quantum Hall channels vulnerability. In turn, this may ease future developments towards precise manipulation of topologically protected edge channels hosted in various types of two-dimensional crystals.

[1] IMCN/NAPS, Université catholique de Louvain (UCLouvain), Louvain-la-Neuve, Belgium. [2] JARA-FIT and 2nd Institute of Physics—RWTH Aachen, Aachen, Germany. [3] Research Center for Functional Materials, National Institute for Materials Science, Tsukuba, Japan. [4] International Center for Materials Nanoarchitectonics, National Institute for Materials Science, Tsukuba, Japan. ✉email: benoit.hackens@uclouvain.be

Quantum Hall edge channels (QHECs), formed as Landau levels (LLs) cross the Fermi energy near the borders of two-dimensional electronic systems (2DESs), are almost ideal one-dimensional systems, where quasiparticle scattering is topologically prohibited[1]. Substantial advances in the manipulation of QHECs in semiconductor-based 2DESs lead to envision new approaches in quantum computing[2–7] and open the way toward electron quantum optics[8]. These breakthroughs require a robust topological protection of QHECs.

Graphene, characterized by the massless nature of its charge carriers, offers even more promising perspectives in terms of QHECs manipulation, thanks to its rich spectrum of relativistic quantum Hall phenomena[9]. In that framework, different strategies relying on QHEC propagation along p-n junctions have already been implemented in this material[10–14]. However, the confinement of charge carriers at graphene borders appears much more difficult to control than in semiconductor-based 2DESs, seriously impairing the topological protection of its QHECs. The explanation lies in different fundamental reasons, including the complex electrostatic screening of the back gate potential related to the presence of fringing fields in most device layouts investigated up to now[15,16], and the difficulty to control defects at the borders of etched graphene[17–20]. The best proof of these detrimental influences is that some fractional quantum Hall signatures visible in extremely clean geometries were only observed in the case of edgeless device layouts such as the Corbino geometry[21,22].

Recently, local probe measurements[23,24], combined with theory[15], led to a revision of the QHECs picture at graphene device edges. Instead of a single type of QHECs propagating along the border in clockwise or anticlockwise fashion as in semiconductor-based 2DES, the new proposed picture involves coexisting downstream and upstream QHECs separated by few 100-nm-wide incompressible (i.e. insulating) strips. Topological breakdown of graphene QHECs would therefore originate from the coupling between up- and downstream QHECs. This coupling has been revealed by Marguerite et al. through scanning probe measurements[24]: on the one side, inelastic scattering was identified as a source of thermal dissipation along up- and downstream QHECs, with no incidence on transport, and on the other side, elastic tunneling was found to cause the coupling between these channels. However, the exact tunneling mechanism, and in particular a clear connection between scanning probe images and macroscopic transport properties, are still lacking.

## Results

**Scanning gate microscopy in the quantum Hall regime**. Here, we use scanning gate microscopy (SGM) to build a full microscopic picture of QHECs topological protection breakdown in graphene. For this purpose, we studied two devices (G1 and G2), consisting in 250-nm-wide encapsulated graphene constrictions as presented in Fig. 1a and functioning only in the p-doped side at high magnetic field (see Supplementary Note 1). Figure 1b displays the longitudinal resistance $R_{xx}$ as a function of back gate voltage $V_{bg}$, showing fingerprints of the QH regime in graphene: $R_{xx}$ vanishes (orange-shaded boxes in Fig. 1b) around the filling factors $\nu = \pm 4(n + 1/2)$, while it is maximal around $\nu = \pm 4n$ (the $n$th LL is aligned with the Fermi energy—see Supplementary Note 1).

In this work, we focus on the transition between the latter two regimes, where $R_{xx}$, while close to zero, exhibits fluctuations (see Supplementary Fig. 6a), signatures of QH topological protection breakdown. Similar fluctuations have been evidenced in transport through constrictions defined in high mobility semiconductor-based 2DESs[25–28]. They have been ascribed to backscattering between QHECs propagating at opposite device edges, occurring

through resonant tunneling via an antidot localized state. This mechanism is particularly effective when the antidot is located in the vicinity of the constriction where QHECs are brought in close proximity.

The antidots locations in real space can be pinpointed thanks to SGM measurements. In SGM, local control over the potential landscape is achieved by electrically polarizing a sharp metallic tip moving in a plane parallel to the device surface. Recording simultaneously $R_{xx}$ as a function of tip position yields SGM maps. In the case of resonant tunneling between QHECs, a moving potential perturbation changes the resonance conditions, turning on and off QHECs backscattering. This yields circular features in SGM resistance maps, centered around the active antidot[28].

In contrast with observations in semiconductor-based 2DEGs, centers of concentric SGM fringes are also located away from the constriction region of our graphene device. SGM images displayed in Fig. 1c–e were obtained at a distance of 500 nm from the constriction, at $V_{bg} = -13$ V, as indicated with an arrow in Fig. 1b, i.e. where the first deviations from $R_{xx} = 0$ emerge, corresponding to the onset of the $\nu = -6$ QH state breakdown. SGM maps allow to pinpoint where the breakdown occurs: indeed, non-zero $R_{xx}$ regions draw sets of concentric rings centered close to the edges, whose number and position evolve with the tip polarization $V_{tip}$ (Fig. 1c–e for sample G1 and Supplementary Fig. 3b–f for sample G2). However, the observation of SGM contrast at large distance from the constriction (about 500 nm in Fig. 1, and a few μm in Supplementary Fig. 3) demonstrates that the constriction does not play a significant role here, which is counter-intuitive in the textbook framework of QH effect in conventional semiconductor-based 2DEGs. In this picture, counterpropagating QHECs run along opposite device edges, and are separated by an insulating bulk region much larger than the tip-induced perturbation. Away from the constriction, the edge states can only circumvent the perturbation and no tip-effect can be expected.

The key missing ingredient in the picture, allowing to solve the puzzling SGM signatures along the devices edges, is electrostatics. Indeed, as predicted by theory[15], inhomogeneous screening of the back gate potential by graphene charge carriers leads to non-monotonic confining potential at the edges (see Supplementary Note 4 for further discussions about the effect that edge impurities could also have on this confining potential). Since LLs follow the same evolution as the potential, as schematically depicted in Fig. 2a, one then expects the presence of both up- and downstream QHECs along the same edge if the Fermi energy crosses twice the same LL. Tunneling between counterpropagating QHECs can be mediated by the presence of localized states associated with antidots, which pin circular QHECs "islands" in-between the QH channels (Fig. 2a). These antidots are at the origin of the characteristic concentric rings of non-zero $R_{xx}$ in Fig. 1c–e. Note that these SGM signatures do not originate from a direct coupling of the counterpropagating QHECs induced by the tip potential alone: this would yield iso-resistance stripes following the edge topography[24,29]. The absence of such stripes in SGM maps (Fig. 1) testifies that the tip perturbation is small enough to avoid inducing direct backscattering.

**Transport through antidots**. Next, we detail how the tip influences tunneling through such an antidot, whose electronic structure has been extensively studied in graphene with scanning tunneling microscopy[30–32]. Antidots host discrete energy levels in the QH regime, whose positions are determined by size confinement in the resultant QHEC island on one hand (quantum contribution) and by Coulomb charging energy on the other hand (electrostatic contribution). A more in-depth discussion on the

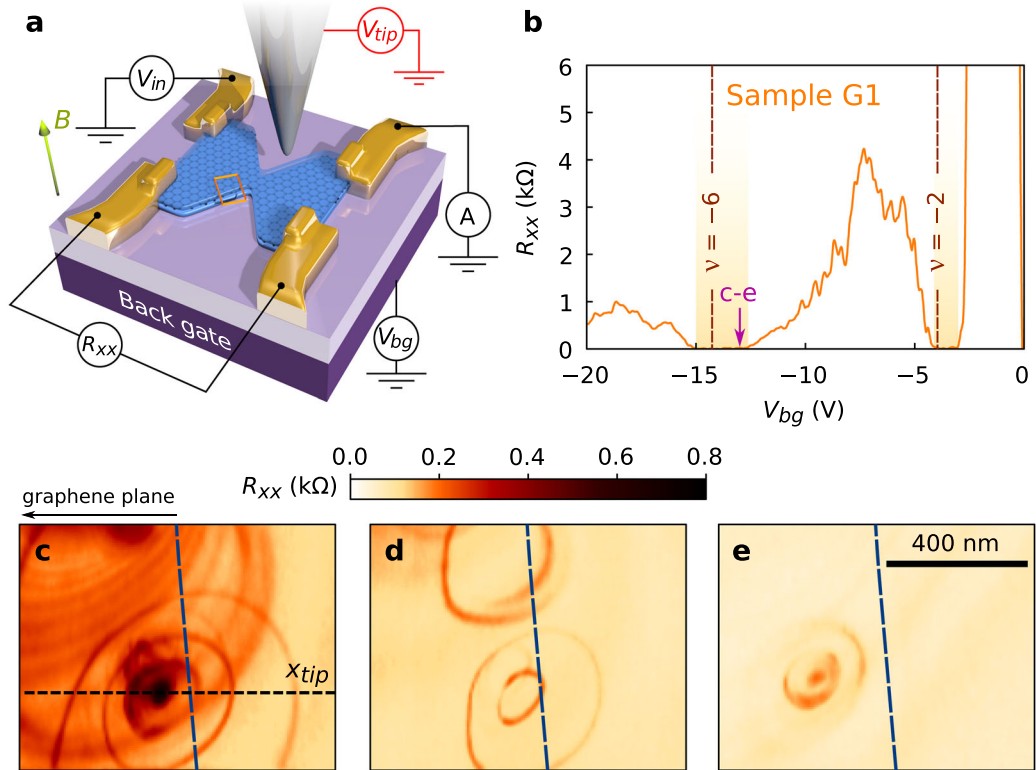

**Fig. 1 Imaging the topological protection breakdown. a** Schematic of the experimental setup. The biased tip can locally change the charge carriers density when applying the voltage $V_{tip}$ and is moved at a distance $d_{tip} \sim 70$ nm above the graphene plane. The global (bulk) charge carrier density in graphene is tuned by the back gate voltage $V_{bg}$. A magnetic field $B$ is applied perpendicularly to the graphene plane. **b** Longitudinal resistance $R_{xx}$ as a function of $V_{bg}$, at $B = 10$ T, measured in sample G1. **c–e** SGM maps of $R_{xx}$ as a function of tip position. The scanning area is sketched by the orange rectangle in **a**, located ~500 nm away from the constriction. The data are recorded with $V_{bg} = -13$ V—arrow in **b**—and $V_{tip} = +3$ V (**c**), 0 V (**d**), and −6 V (**e**).

different contributions is given in Supplementary Note 2.2. Discrete energy levels are shifted under the tip-induced local modification of potential landscape, as sketched in Fig. 2b, c. The high $R_{xx}$ rings in Fig. 1c–e are the loci of tip positions leading to an alignment between one of the antidot's discrete energy levels and the QHECs potential (Fig. 2c), whereas low $R_{xx}$ between the rings corresponds to Coulomb blockade[33,34] (Fig. 2b). This picture is confirmed by the emergence of Coulomb diamonds in scanning gate spectroscopy[35]: applying a DC bias between source and drain allows to overcome Coulomb blockade as soon as the source-drain energy windows overlaps a localized state energy (see Supplementary Note 3). In this framework, the position of the antidot corresponds to the center of the Coulomb rings (at low $V_{tip}$, screening effects can however distort and shift Coulomb rings, as discussed in Supplementary Note 2). Based on Fig. 1c–e, we pinpoint antidots positions at a distance between 50 and 150 nm from sample G1 boundaries. This is in agreement with the estimated upstream QHEC position extracted from recent local probe results[23,24].

A fundamental question emerging at this point concerns the origin of the observed antidots. Atomic defects at the edges of graphene have often been invoked as source of perturbation for charge transport. However, if they were involved in the present case, it would remain to explain how they could yield potential landscapes similar to the one presented in Fig. 2a, with a potential extremum located 50–150 nm from the edge. More realistically, such potential landscape could originate from two known possible sources: (1) nanoscale random strain fluctuations, known to induce charge density inhomogeneities in graphene[36] (2) remote charged impurities in the dielectric hBN layer[37]. Both sources lead to local variations of Dirac point energies (typically

~50–100 meV at $B = 0$ T, over typical distances ~50–100 nm[38]), probably ubiquitous in all hBN/graphene/hBN heterostructures. It is noteworthy that a potential fluctuation giving rise to an antidot on the p-doped side would yield a dot on the n-doped side. While our experiment does not allow to discriminate between strain- or impurity-induced potential fluctuations, it provides data on antidots distance from device borders, as well as on their spatial distribution along the borders of graphene devices : the typical distance between neighboring antidots is in the range 100–500 nm, from data in Fig. 1c–e and Supplementary Fig. 3, i.e. compatible with data from ref. [38]. Since the tip-induced potential perturbation extends beyond 500 nm, Coulomb rings originating from remote antidots can superimpose, as shown on Fig. 1c–e and Supplementary Figs. 3.

**Back gate and tip control of antidots**. The spatial locations of the antidots being unveiled, we now examine how their signatures emerge and evolve as a function of $V_{bg}$. For this purpose, we scan the tip across one of the antidots as indicated in Fig. 3a (the scan area in Fig. 3a corresponds to the red rectangle in Fig. 3b) and plot in Fig. 3c the SGM line profile as a function of $V_{bg}$ in the vicinity of $\nu = -6$ for a constant $V_{tip}$ (see Supplementary Note 2.3). It is well known from earlier SGM experiments on Coulomb blockaded islands that such a plot allows to infer the tip potential perturbation from the $V_{bg}$-shift of Coulomb blockade resonances[33,39]. Coulomb resonances undergo a Lorentzian evolution, as shown by the fits in Fig. 3c, as expected for a tip-induced potential perturbation (see Supplementary Note 2.3). Examining Fig. 3a, c, d together, one can get the full picture of the fate of Coulomb resonances associated with antidots: Fig. 3c, d evidence that peaks identified by the red and blue dashed lines

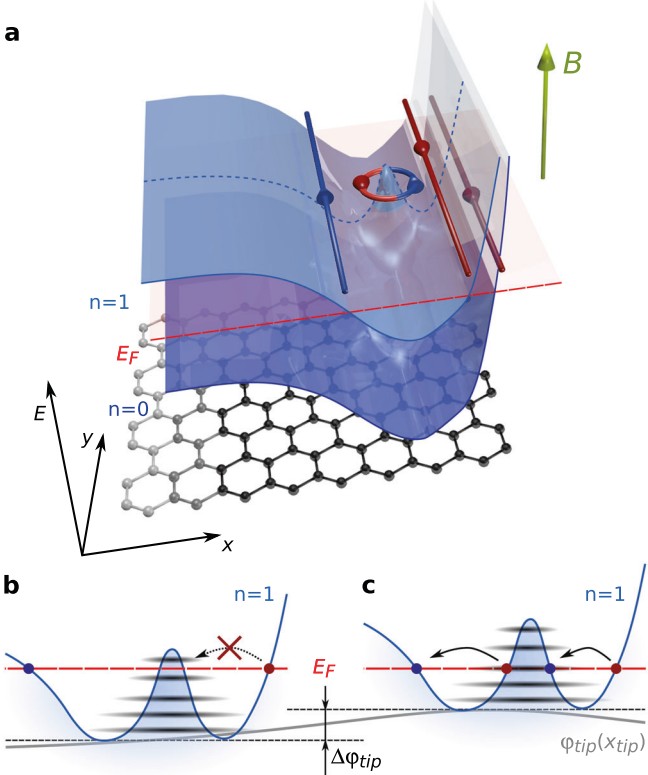

**Fig. 2 Artist' view of QHECs at graphene edge. a** The two lowest LLs arising due to the perpendicular magnetic field (green arrow) are represented as blue semi-transparent surfaces. Because of electrostatics at the graphene edge (on the right side), they are bent and the $n = 1$ LL crosses twice the Fermi energy $E_F$ (red plane) yielding two downstream QHECs (in red) and to an upstream QHEC (in blue). An antidot is located between the counterpropagating QHECs and pins a QHEC island. **b**, **c** Line profile across the QHEC island (blue dotted line in **a**) for the $n = 1$ LL (electron charge carriers). Discrete energy levels are represented in black. The tip-induced potential $\varphi_{tip}(x_{tip})$ (gray line) tunes discrete energy levels positions with respect to $E_F$ when varying the tip position $x_{tip}$. When $E_F$ lies between two discrete energy levels, transport is not allowed via the QHEC island (**b**) whereas when a discrete energy level is aligned with $E_F$, charge carriers can tunnel between the counterpropagating QHECs (red and blue dots) through the QHEC island (**c**).

undergo a parallel evolution with the approaching tip perturbation, and are therefore associated with the same antidot, whose location is clearly identified in the SGM map in Fig. 3a. Importantly, the Coulomb resonances are also observed when the tip is far away from the device edges which means that the tunneling through the antidot is not necessarily triggered by the tip potential. Indeed, the Coulomb resonance signatures can be tuned by $V_{bg}$ as shown in Fig. 3d.

A more intriguing behavior is also revealed for the resonance highlighted by the red dashed lines in Fig. 3c: below $V_{bg} < -21.5$ V (black dashed line), signatures of this resonance vanish. This $V_{bg}$ threshold is independent of $V_{tip}$ as demonstrated in Supplementary Fig. 4. We propose the following picture to understand this phenomenon. Resonances are only visible provided that (1) a discrete state associated with an antidot is tunnel-coupled to up- and downstream QHECs as depicted in Fig. 3e, g and (2) the upstream QHEC allows charge carriers to be sent back to the injection contact. Varying $V_{bg}$ has a strong influence on the position of the upstream QHEC (blue in Fig. 3g, h). As soon as the tunnel coupling becomes too small as illustrated in Fig. 3f, h or the upstream QHEC is no more connected to the injection contact,

backscattering through the antidot is no longer effective and the resonance signature disappears. These data are crucial as they confirm the presence and the contribution of forward- and backward-propagating QH states at the device border.

The coupling between the upstream QHEC and the injection contact is essential to understand the link between the QHECs structure and the filling factor deduced from transport measurements. Considering that this coupling is not perfect, the apparent filling factor is not defined by the bulk (dark purple in Fig. 3b) but rather by the incompressible region between the up- and downstream QHECs (light purple in Fig. 3b). In Fig. 3e, f, the filling factor therefore takes a value $\nu \sim -6$ even if the bulk filling factor is $-2$. We have further discussed the coupling between QHECs and the contacts in graphene samples in[40].

Another way to tune the position and configuration of QHECs, but at the local scale, consists in varying both tip voltage and position. This is realized in Fig. 4a showing the evolution of $R_{xx}$ when scanning the tip along the dashed line in Fig. 1d and varying $V_{tip}$. The different visible resonances corresponding to the same antidot undergo parabolic evolution with $V_{tip}$ as expected for localized states[39]. At low $V_{tip}$, these resonances are separated by $R_{xx} \sim 0$ regions (corresponding to Coulomb blockade), while a finite $R_{xx}$ region (in dark in Fig. 4a) is reached at larger positive $V_{tip}$. This evolution is also clearly visible in Fig. 4b showing $R_{xx}$ versus the maximum tip-induced decrease in hole density $|\Delta n_{tip}|$ deduced from $V_{tip}$ (see Supplementary Note 5), for a fixed $x_{tip}$ (with the tip on top of the antidot—black dotted line in Fig. 4a). At lower tip perturbation, transport is determined by tunneling through the antidot as discussed above (left inset of Fig. 4b). As the tip-induced perturbation increases, the antidot grows and merges with up- or downstream QHECs. The confinement of charge carriers in the antidot is then suppressed and the backscattering is only induced by the coupling between the counterpropagating QHECs, as depicted in the right inset of Fig. 4b and further detailed in Supplementary Note 6.

**Simulations**. Tight-binding simulations reproduce the observed phenomenology and provide further insights in the underlying physics through real space images of the local current density (JDOS) in the different backscattering regimes. Using the KWANT package[41] (see Supplementary Note 7), we model one edge of the device as a 150-nm-wide graphene ribbon represented in Fig. 4d where the colors correspond to the onsite potential landscape. In our simulations, we focus on a single side of the device, and neglect the bulk region contribution. The antidot potential is positioned close to the center of Fig. 4d. In this geometrical configuration, counterpropagating QHECs (straight arrows in Fig. 4e) encompass the QHEC island (curved arrows in Fig. 4e) for the Fermi energy corresponding to the red dashed line of Fig. 4e. The tip potential shifts the relative position of the LLs with respect to the Fermi energy, thereby tuning the distance and coupling between the QHECs and the antidot.

Noteworthy, we observe a striking qualitative correspondence between the measured (Fig. 4b) and simulated (Fig. 4c) longitudinal resistance as a function of $|\Delta n_{tip}|$: at low $|\Delta n_{tip}|$, finite $R_{xx}$ peaks are separated by $R_{xx} \sim 0$ states and at larger $|\Delta n_{tip}|$, $R_{xx}$ remains finite. The $|\Delta n_{tip}|$ scale (distance between the peaks) depends mainly on the size of the considered antidot as well as on Coulomb interactions, not captured in our simulations. Since all the parameters vary among the antidots, the comparison between experimental and simulated typical $|\Delta n_{tip}|$ scales will remain qualitative. The sequence of JDOS maps shown in Fig. 4f, g provides a real space illustration of the peaks' origin. Comparing Fig. 4f, g, corresponding respectively to finite and zero $R_{xx}$ (see Fig. 4c), we observe that, while in both cases the antidot is

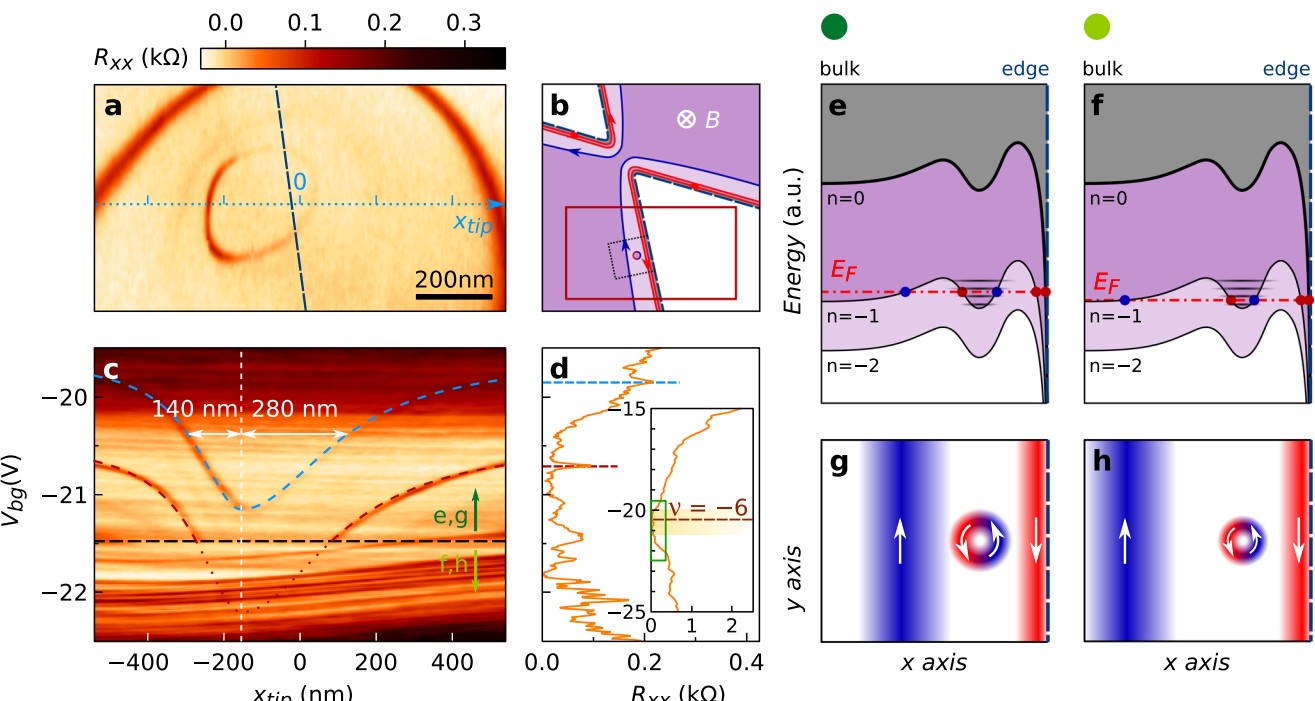

**Fig. 3 Coupling counterpropagating QHECs via an antidot.** The data are obtained in sample G1 for $V_{tip} = 0$ V and $B = 14$ T. **a** SGM map obtained at $V_{bg} = -20.85$ V, by scanning the tip inside the red rectangle indicated in the schematic picture of the device sketched in **b**. **b** The QHECs are represented by red (downstream) and blue (upstream) continuous lines, and dashed line delineate the constriction. **c** $R_{xx}$ recorded as a function of $V_{bg}$ and the tip position $x_{tip}$, along the light blue dotted line in **a**. The resonances signatures (highlighted with the red and blue dashed lines) allow to measure the tip-induced potential variation at the QHEC island location as a function of $x_{tip}$. Blue and red dashed lines are fits obtained with two merged half-Lorentzian functions. Above graphene, the half width at half maximum is 140 nm whereas it is 280 nm when the tip is above the etched area. The black dashed line indicates the $V_{bg}$ limit beyond which one of the resonances disappears. **d** Longitudinal resistance $R_{xx}$ as a function of $V_{bg}$ around $\nu = -6$—zoom on the green rectangle of the inset. Schematics of the three lowest LLs, following the potential profile (thicker line) along $x_{tip}$-axis in map **c** for $V_{bg} > -21.5$ V (**e**) and $V_{bg} < -21.5$ V (**f**). **g** Schematic of the QHECs in real space, at the Fermi energy indicated by the red dash-dotted line in **e** (downstream in red and upstream in blue). The circular QHEC is pinned at the location of the antidot. **h** Real space schematics of QHECs corresponding to Fermi energy indicated by the red dash-dotted line in **g**, where the upstream channel vanishes.

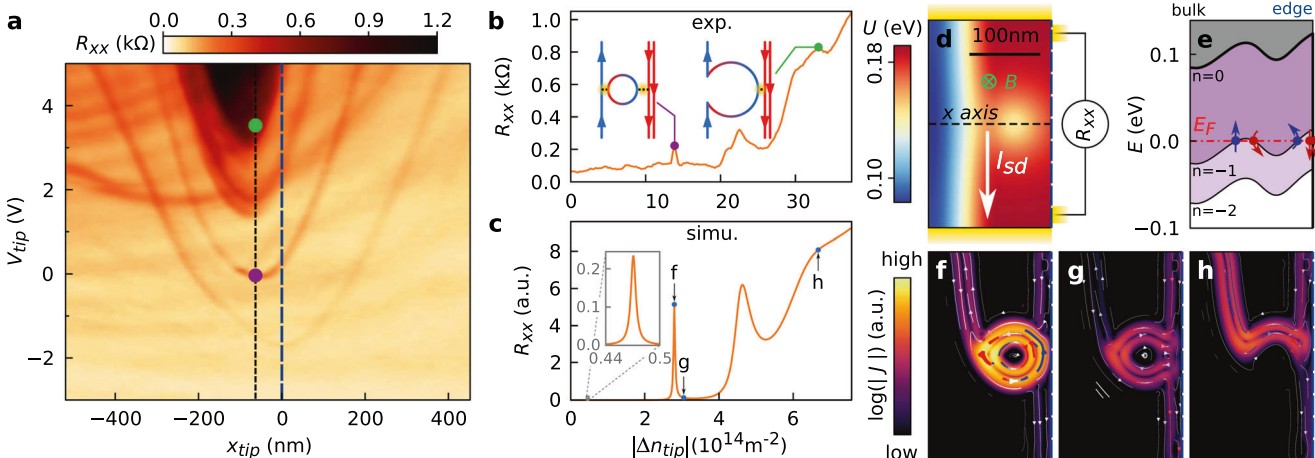

**Fig. 4 Tip-controlled tuning of transport through a QHEC island. a** Evolution of $R_{xx}$ as a function of $V_{tip}$ (measured along the black dotted line in Fig. 1c at $V_{bg} = -13$ V and $B = 12$ T). **b** $R_{xx}$ profile for $x_{tip} \sim -60$ nm (tip on top of the antidot, i.e. along the black dotted line in **a**). $V_{tip}$ has been converted in the maximum tip-induced hole density decrease $|\Delta n_{tip}|$. **c** Simulations of $R_{xx}$ as a function of $|\Delta n_{tip}|$ at the lower edge of $\nu = -6$ plateau, at $B = 12$ T. **d** Scheme of the simulated system, with colors corresponding to the onsite potential landscape. The antidot corresponds to the circular region where the potential is lower, centered at 45 nm from the graphene edge. The four leads required to compute $R_{xx}$ are represented in yellow. **e** Profile of the three lowest LLs ($n = 0, -1, -2$) along the black dashed line in **d**. This graph is similar to Fig. 3f, g, except for the infinitely sharp confinement potential at the edge (right side of the figure) in the simulation, which yields two downstream QHECs (red straight arrows). **f**–**h** Simulated maps of current density (JDOS) obtained for the three $\Delta n_{tip}$ values indicated with arrows in **c**. On top of the $R_{xx}$ peak (**f**), the JDOS around the antidot is maximal compared to the situation of zero $R_{xx}$ (**g**). The high JDOS in the antidot highlights that the resonance condition is reached. **h** The region of finite $R_{xx}$ in **c** corresponds to direct backscattering of QHECs. The colored arrows indicate the direction of the local current density.

coupled to downstream QH channel (right of the figures), current through the antidot is significantly larger in the case of Fig. 4f (as indicated by the brighter contrast in log scale at the antidot position). Coupling between up- and downstream QH channels is therefore much more efficient, yielding finite $R_{xx}$. At much higher $|\Delta n_{tip}|$ (Fig. 4h), the JDOS map reveals that the raised antidot potential results in the merging of the antidot with the upstream QHEC, confirming the schematic picture sketched in the right inset of Fig. 4b.

## Discussion

Put together, our data shed a new light on the combined role of electrostatics (fringing fields or charged impurities) and antidots at graphene edges in QH breakdown. Both ingredients are likely ubiquitous in most graphene-based heterostructures studied up to now, but with variations in the importance of the different contributions. Indeed, fringing fields become much weaker when the gate is placed closer to graphene, for example when a graphite back gate is used below hBN. Furthermore, charged impurities at hBN etched edges depend on the etching recipe, and Dirac point inhomogeneites may be more or less pronounced depending on strain accumulated in the layers or on the quality of hBN.

SGM data obtained at high magnetic field allow to get precise information on active antidot locations (distance from the border, and distribution along the border), putting constraints on their possible origin. The fine control over antidot size and coupling to QHECs provided by the tip and back gate voltages was shown here to be the key to disentangle the complexity of the QH effect phenomenology in graphene. It allows to image and tune antidot-mediated QH effect breakdown, which constitutes a prerequisite toward advanced control and manipulation of QHECs in more complex devices such as QH interferometers. These findings are indeed relevant, for example, in the case of p-n junction-based interferometers where semi-reflecting mirrors are defined at the edges[11,14]. Noteworthy, the main outcome of this work, that full control over topological edge states in graphene will only be provided through meticulous engineering of electrostatic landscape at device borders, can also be transposed to other types of 2D crystal-based devices hosting topologically protected edge states.

## Methods

**Samples fabrication**. Sample G1, depicted in Fig. 1a, consists in a graphene flake encapsulated between two hBN layers (20 nm thick for the top layer and 30 nm thick for the bottom layer) using dry transfer techniques and deposited on a doped Si wafer covered by a 300-nm-thick SiO$_2$ layer. A 250-nm-wide constriction shape has been lithographically defined, similarly to[18]. The four contacts allow to measure the longitudinal resistance $R_{xx}$.

Sample G2, depicted in Supplementary Fig. 2a, has been built with the same processes as sample G1 and with the same hBN layers thicknesses. The constriction has the same width. The major difference with sample G1 lies in the presence of six contacts, allowing to measure the Hall resistance $R_{xy}$ in addition to $R_{xx}$.

**Measurements technique**. The sample has been anchored to the mixing chamber of a dilution refrigerator whose base temperature is 100 mK and a magnetic field $B$ up to 14 T has been applied perpendicularly to the graphene plane. Electrical signals have been recorded using a classical lock-in technique at a frequency of 77 Hz. The longitudinal resistance is obtained from a four probe measurement to avoid the contribution from contacts resistance. Charge carriers type and density can be tuned by changing the back gate voltage $V_{bg}$.

The local gate used for SGM characterization consists in a commercial metal-coated AFM tip glued on a tuning fork whose resonance frequency is $f \sim 32$ kHz. The tip is electrically contacted so that a voltage $V_{tip}$ can be applied on it. The tip can be moved in $x$, $y$, $z$ directions thanks to piezo scanners. After scanning the surface in topography mode, the distance $d_{tip}$ between the tip and the graphene plane can be fixed. Applying the bias $V_{tip}$ introduces an electrostatic perturbation for conduction electrons. The conductance can then be recorded for each tip position, yielding a SGM map.

**Simulations**. Tight-binding simulations have been performed using the KWANT package[41]. We modeled one edge of the device, neglecting the bulk region contribution (see Supplementary Fig. 8), as a 150-nm-wide graphene ribbon represented in Fig. 4d where the colors correspond to the onsite potential landscape. This potential is asymmetric along the $x$-axis, resulting in the spatial profile for the LLs shown as black lines in Fig. 4e. Their shape matches the qualitative picture given in Fig. 3e, g for the energy levels' evolution close to the edge of the graphene device. Note that the confinement is infinitely sharp in the simulation at the device border (right side of Fig. 4d, e), contrary to the smoother evolution schematically depicted in Fig. 3e, g, without consequence on the qualitative correspondence between simulation and experimental results. Finally, the antidot potential has been modeled by a Gaussian function and is positioned at 45 nm from the edge.

To decrease computation time, a scaling factor of four, without incidence on the output results, was applied to the real lattice parameter of graphene (the interatomic distance is $a = 4 \times 1.42$ Å and the hopping parameter is $t = 2.7/4$ eV). More details on simulations are available in Supplementary Note 7.

## Data availability

The raw experimental data generated in this study have been deposited in the following database: https://doi.org/10.14428/DVN/SFT7SF.

## Code availability

The tight-binding codes used to produce the simulations presented in this article are available at https://doi.org/10.5281/zenodo.4945102.

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

## Acknowledgements

The authors thank Vincent Bayot and Markus Morgenstern for their valuable comments and corrections. N.M. thanks Sébastien Toussaint for enriching discussions. B.B. thanks Julien Renard for enlightening discussions. This work was partly funded by the Federation Wallonie-Bruxelles through the ARC Grant No. 16/21-077, by the F.R.S-FNRS through the Grant No. J008019F, and from the European Union's Horizon 2020 Research and Innovation program (Core 1 No. 696656 and Core 2 No. 785219). This work was also partly supported by the FLAG-ERA grant TATTOOS, through F.R.S.-FNRS PINT-MULTI grant No. R 8010.19. Computational resources have been provided by the Consortium des Equipements de Calcul Intensif (CECI), funded by the Fonds de la Recherche Scientifique de Belgique (F.R.S.-FNRS) under Grant No. 2.5020.11 and by the Walloon Region. B.B. (research assistant), B.H. (research associate), and N.M. (FRIA fellowship) acknowledge financial support from the F.R.S.-FNRS of Belgium. Support by the Helmholtz Nanoelectronic Facility (HNF), the EU ITN SPINOGRAPH and the DFG (SPP-1459) is gratefully acknowledged. K.W. and T.T. acknowledge support from the Elemental Strategy Initiative conducted by the MEXT, Japan, Grant Number JPMXP0112101001, JSPS KAKENHI Grant Numbers JP20H00354 and the CREST (JPMJCR15F3), JST.

## Author contributions

N.M. performed the experimental measurements, with the assistance of B.B. N.M. analyzed the data. N.M. performed the tight-binding simulations. S.S. fabricated the samples. K.W. and T.T. synthesized the hBN crystals. N.M. and B.H. wrote the paper. B.H. and C.S. supervised the collaboration. All authors discussed the results and commented on the manuscript.

## Competing interests

The authors declare no competing interests.
