## [Peer Review File · Nature Communications]

REVIEWER COMMENTS

Reviewer #1 (Remarks to the Author):

The manuscript by Moreau and co-workers reports scanning gate microscopy of graphene in the quantum Hall regime. They focus on the scattering between counter-flowing edge channels of non-topological origin that have recently been revealed to be the source of quantum Hall effect breakdown in graphene. They observe concentric rings near the sample edge in their scanning gate image, which they explain as indicating backscattering mediated by resonant levels formed around an antidot. Dependence on back gate voltage, tip position and tip voltage, together with tight-binding transport simulation, are shown to be broadly consistent with this picture.

In my view, the manuscript is clearly written, the experimental data are of high quality, and the analysis and interpretation appear sound. The microscopic mechanism of topological protection breakdown in graphene quantum Hall effects is a hot issue, which makes this manuscript potentially important and a candidate for publication in Nature Communications. Aside from some technical issues described below, the important question that makes me reserve recommendation is the generality of the reported results. Although the antidot picture seems successful in explaining the scanning gate data of the authors, it is not clear whether the same mechanism is relevant to the majority of reported experiments on graphene quantum Hall effects. To appropriately evaluate the importance of this work, I would like to see a revised manuscript where this point is better clarified. In addition, the technical issues described below should also be addressed.

1) In the inset of Fig. 3d, the filling factor $\nu = -6$ is located at about $V_{bg} = -20.4$ V. However, the corresponding Landau level profile and the position of the Fermi level shown in Fig. 3e suggest that the bulk filling factor is -2 instead. The same applies to Fig. 4e. If I understand correctly, this is because, due to the hole accumulation at the edge, the filling factor near the edge is higher (for holes) than in the bulk and the Hall resistance is quantized reflecting the filling factor of the incompressible stripe along the edge, as proposed by Cui et al. in Ref. 22. Most of the readers are not familiar with this picture, so I think it needs to be explained more explicitly. In particular, the authors should specify what filling factor ν represents. Another suggestion would be to indicate the values of local filling factors “-2” and “-6” in Fig. 3b.

2) I am confused about the effect of scattering between counter-propagating edge channels on R_{xx} . In the situation shown in the right inset of Fig. 4b, the antidot has become so large that the counter-propagating edge channels are fully reflected. In a naïve picture, this is equivalent to having no counter-propagating edge channels, meaning that R_{xx} should be zero. This is the case when all the edge channels, including the upstream one shown in blue, are connected to ohmic contacts. In

contrast, in the experiment shown in Fig. 4b, R_{xx} is finite when the counter-propagating edge channels are fully reflected. A similar behavior is reproduced in a tight-binding simulation. However, the correspondence between the geometry used for the simulation (Fig. 4d and Fig. S8c) and the actual experimental configuration (Fig. 1a) is unclear. Although supplementary Fig. S8c explains this point, it is not clear enough. As I understand, the picture of quantum Hall effects in Ref. 22 requires that the upstream edge channel is decoupled from ohmic contacts. Then I wonder how one can simulate transport in such a situation. Perhaps, adding ohmic contacts, voltage source, and the ground to Fig. 4d and Fig. S8c might help. (Also, the ohmic contacts in Fig. 4d should be indicated by color distinct from red and blue.)

3) The scanning gate images reported in this manuscript show only features related to antidots. However, in principle, just a tip-induced potential can also affect the scattering between counter-propagating edge channels, which should emerge in scanning gate images as in Ref. 23. Why are only features due to antidots visible? Is it because the authors are focusing on the filling factor region where the incompressible strip is very wide? The authors should elaborate on this point, together with the connection to other reported experiments.

4) (optional) The authors emphasize that contrast in scanning gate image appears at a large distance from the constriction to highlight their findings. However, it appears to me that the explanation starting from constriction makes it more difficult instead to understand what the authors are actually looking at. Firstly, this is because the mechanism for the R_{xx} increase is different for the backscattering at the constriction and that between counter-propagating edge channels on one side. Secondly, from Fig. 1a the readers would expect that something is happening at the constriction, but actually, it does not play a significant role, as the authors state. (It is not clear from reading whether the constriction plays any role at all, though. This should be clarified. Perhaps it does not play any role here.) Thirdly, the possibility of non-topological counter-propagating edge channels is already mentioned in the introduction. So, the authors do not need to start from “the textbook framework of QH effect.”

5) It is hard to tell what the blue and purple circles above Figs. 3e, f represent. I guess these correspond to range of V_{bg} shown by the blue and purple arrows in Fig. 3c. Since these colors are similar to the blue and red colors representing the upstream and downstream edge channels, alternative colors should be used.

Reviewer #2 (Remarks to the Author):

The authors employ a scanning gate microscope to study the graphene transport in the quantum Hall regime. They find that the QH transport can be perturbed by the tip, and they explain the behavior by considering an antidot near the graphene edge that could mediate backscatterings between counter-propagating QH edge channels formed due to electrostatic potential. While I appreciate the interesting data on probing such impurities, I have a few concerns about the general implications of these results. I am not convinced that this paper meets the broad audience criterion for publication in Nature Comm.

1. The presence of such local potential perturbation in graphene, whether it is due to defects or charge impurities, is not surprising. The question is how they would affect the QH transport. The paper's title says "Upstream modes and antidots poison graphene quantum Hall effect", but it seems that such poisoning occurs only when a tip is near the edge. What would be the effect of such antidot on QH transport without a tip present? I think this is an important question for the broad audience to know.

2. In the schematics in Fig.2, a current loop is drawn around the antidot. I am a little confused about this. Does the loop represent the QH edge channels in the antidot? If so, the scenario in Fig. 2b should also cause backscatterings, through the edge channels around the loop.

3. It is known that in the QH regime, electrostatics can cause reconstruction and form incompressible regions (PRB 46, 4026 (1992)). What is the role of such incompressible region in the present case?

4. The tip voltage can induce a local charge accumulation and create incompressible rings around such charge bubble in the QH regime, as demonstrated by a previous SPM experiment (PRL 95, 136804 (2005)). The authors should consider if and how this effect affects the analysis.

5. I think Fig. 3e and 3f are misleading. These two are supposed to describe the scenario around $\nu=-6$ plateau. In Fig. 3e, the bulk E_f is between $n=0$ and $n=-1$, which should correspond to $\nu=-2$ plateau. In the QH regime, the chemical potential is not linear in terms of density or gate. Also, what are the tilted lines in Fig. 3c at $V_{bg} < -21.5V$? Why do these features depend on tip position?

6. In Fig.4a, what sets the polarity of the observed parabola features? I suppose similar parabola facing downward can be observed on the negative tip voltage side but perhaps at a different back gate? The work function difference of 3V between tip and graphene seems to be rather large. What is the material of the metal on the tip?

Reviewer #3 (Remarks to the Author):

The goal of the paper "Upstream modes and antidots poison graphene quantum Hall effect" is the experimental investigation of the instability of Quantum Hall edge transport in realistic graphene samples. Authors defend the scenario there electrostatic accumulation of charges at the edges of graphene flake results in existence of both downstream and upstream counter-propagating edge channels in the integer Quantum Hall effect regime. Then authors suggest that special impurities "antidots" which happen to exist near the edges of graphene would lead to electron's scattering into the upstream channels, breaking effectively the topological protection of the current.

Using the scanning gate microscope authors were able to identify such antidots and to measure directly the current distribution inside the impurity/antidot and in the both downstream and upstream channels.

To my opinion this is an interesting result, which deserves to be published.

Below I give a list of questions, which I suggest to consider before publication.

1. Authors say (page 3 right bottom) that the distance between antidots is 100-500 nm. Where this comes from? Is it the bulk distance between randomly distributed antidots, or the typical distance between the impurities along the edge?
2. Am I right that all the experiments were performed in the hall doped graphene (negative bias in Fig. 1b)? Why? And may authors say that explicitly?
3. Why the impurities are called antidots? In usual semiconductors with parabolic dispersion one indeed needs antidots for similar effects. But in graphene with its Dirac spectrum, what is a quantum dot for electrons became an antidot for holes.

4. I think, there is a related physical question. Authors confess, that the exact nature of impurities-antidots is unknown. One possibility is that they are due to the charged impurities in hBN layer. In that case they may have the charge of always the same sign. Is that so? From the first experiments in graphene the Quantum Hall effect was observed for both Fermi energy below and above the Dirac point. But the electrostatic charge accumulated at the edges would have opposite sign in these two cases, while the impurity potential would stay the same. Would that make the effect particle-hole asymmetric? Can this be seen and used to support the picture suggested by the authors?

Reviewer #1 (Remarks to the Author):

The manuscript by Moreau and co-workers reports scanning gate microscopy of graphene in the quantum Hall regime. They focus on the scattering between counter-flowing edge channels of non-topological origin that have recently been revealed to be the source of quantum Hall effect breakdown in graphene. They observe concentric rings near the sample edge in their scanning gate image, which they explain as indicating backscattering mediated by resonant levels formed around an antidot. Dependence on back gate voltage, tip position and tip voltage, together with tight-binding transport simulation, are shown to be broadly consistent with this picture.

In my view, the manuscript is clearly written, the experimental data are of high quality, and the analysis and interpretation appear sound. The microscopic mechanism of topological protection breakdown in graphene quantum Hall effects is a hot issue, which makes this manuscript potentially important and a candidate for publication in Nature Communications. Aside from some technical issues described below, the important question that makes me reserve recommendation is the generality of the reported results. Although the antidot picture seems successful in explaining the scanning gate data of the authors, it is not clear whether the same mechanism is relevant to the majority of reported experiments on graphene quantum Hall effects. To appropriately evaluate the importance of this work, I would like to see a revised manuscript where this point is better clarified. In addition, the technical issues described below should also be addressed.

We thank Reviewer #1 for his enthusiastic general comments about the quality of data and analysis. Concerning the technical issues, we refer to the answers to each specific point listed hereafter, and the corresponding modifications.

We fully agree with the importance of the Reviewer #1's main concern, about the generality of the reported results and in particular of the antidot-mediated breakdown of the graphene QHE. Two ingredients are necessary to observe this type of breakdown: 1) the presence of local variations of the electrostatic potential leading to the local maxima or minima that can "pin" the antidots, and 2) the upstream modes. Each ingredient is very likely to play a role in most graphene devices studied up to now:

- 1) scanning tunneling microscopy (STM) data obtained on graphene on top of hBN reveal local inhomogeneities in the electronic local density of states, over typical lengthscales ~ 50 - 100 nm and corresponding to variations of ~ 50 - 100 meV in Dirac point energies [J. Xue et al., Nature Mat. 10, 282 (2011)]. These numbers are compatible with our observations on antidots, so they reinforce the possibility to generalize our results to graphene/hBN heterostructures deposited on Si/SiO₂ substrates. It is however likely that these numbers vary with the thickness of the bottom hBN layer (i.e. distance between graphene and Si substrate), which means that the importance of antidot-mediated breakdown may vary from one sample to another. The presence of a graphite backgate is also known to completely change the electrostatic picture and yield much smaller variations, compared with bare Si/SiO₂ substrates.
- 2) Upstream modes have already been invoked in [Marguerite, A. et al. Nature 575, 628 (2019)]. The debate is not closed as whether they originate from fringing fields related to the back gate voltage, or from charged impurities located at the borders of the etched edge of the heterostructures. The importance of fringing fields varies a lot as a function of the distance to the gate, the presence of local metallic gates on top of the heterostructures, etc. If fringing fields are at the origin of upstream modes, our conclusions apply to the case of graphene/hBN heterostructures deposited on top of Si/SiO₂ substrate, acting as a global back gate. The latter configuration was the most frequent one in experimental works published in the field during the last ten years. In the case of graphite back gate however, the picture is most probably not the same. If impurities are at the origin of upstream modes, our conclusions also apply to most etched heterostructures, since etching recipes are very similar

in most research groups. In the case of structures limited by “natural edges” of the graphene and hBN flakes, the picture is likely to be different. We emphasize again that the mechanisms described in the manuscript likely apply to topological breakdown in other 2D materials.

We have modified the text to incorporate this discussion, in particular

- in page 4 “[...] Both sources lead to local variations of Dirac point energies (typically ~ 50 - 100 meV at $B=0$ T, over typical distances ~ 50 - 100 nm [J. Xue et al., Nature Mat. 10, 282 (2011)]), probably ubiquitous in all hBN/graphene/hBN heterostructures. While our experiment does not allow to discriminate between strain- or impurity-induced potential fluctuations, it provides data on antidots distance from device borders, as well as on their spatial distribution along the borders of graphene devices : the typical distance between neighboring antidots is in the range 100 - 500 nm, from data in Figs. 1c-e and S3, i.e. compatible with data from ref. [J. Xue et al., Nature Mat. 10, 282 (2011)]. Since the tip-induced potential perturbation extends beyond 500 nm, Coulomb rings originating from remote antidots can superimpose, as shown on Figs. 1c-e and supplementary Figs. S3.”
- just before the last paragraph of the paper “Put together, our data shed a new light on the combined role of electrostatics (fringing fields or charged impurities) and antidots at graphene edges in QH breakdown. Both ingredients are likely ubiquitous in most graphene-based heterostructures studied up to now, but with variations in the importance of the different contributions. Indeed, fringing fields become much weaker when the gate is placed closer to graphene, for example when a graphite backgate is used below hBN. Furthermore, charged impurities at hBN etched edges depend on the etching recipe, and Dirac point inhomogeneities may be more or less pronounced depending on strain accumulated in the layers or on the quality of hBN.

Furthermore, the mechanisms described in the manuscript are likely to play a role in several reported experiments related to quantum Hall interferometry. For example, it has been shown that the mixing between QHECs in a pn junction-based interferometer occurs exclusively along the device edges [Wei, D. S. et al. Science Advances 3 (2017), Jo, M. et al. arXiv (2020) 2011.04958]. We have added a sentence about this in the conclusion :

"These findings are indeed relevant, for example, in the case of pn junction-based interferometers where semi-reflecting mirrors are defined at the edges [Wei, D. S. et al. Science Advances 3 (2017), Jo, M. et al. arXiv (2020) 2011.04958]."

1) In the inset of Fig. 3d, the filling factor $\nu = -6$ is located at about $V_{bg} = -20.4$ V. However, the corresponding Landau level profile and the position of the Fermi level shown in Fig. 3e suggest that the bulk filling factor is -2 instead. The same applies to Fig. 4e. If I understand correctly, this is because, due to the hole accumulation at the edge, the filling factor near the edge is higher (for holes) than in the bulk and the Hall resistance is quantized reflecting the filling factor of the incompressible stripe along the edge, as proposed by Cui et al. in Ref. 22. Most of the readers are not familiar with this picture, so I think it needs to be explained more explicitly. In particular, the authors should specify what filling factor ν represents. Another suggestion would be to indicate the values of local filling factors “ -2 ” and “ -6 ” in Fig. 3b.

We thank Reviewer #1 for this pertinent remark. We have clarified this point for readers not familiar with this picture in an additional paragraph (page 5): “The coupling between the upstream QHEC and the injection contact is essential to understand the link between the QHECs structure and the filling factor deduced from transport measurements. Considering that this coupling is not perfect, the apparent filling factor is not defined by the bulk (dark purple in Fig. 3b) but rather by the incompressible region between the up- and downstream QHECs (light purple in Fig. 3b). In Figs. 3e,f, the filling factor therefore takes a value $\nu \sim -6$ even if the bulk filling factor is -2 . We have

further discussed the coupling between QHECs and the contacts in graphene samples in [arXiv:2103.10331].”

As indicated in this new paragraph, we realized that both the different filling factor close to the edges and the coupling between upstream modes and contacts is crucial in our picture, and we have further developed this point in a paper recently deposited on arXiv and submitted for publication [arXiv:2103.10331].

2) I am confused about the effect of scattering between counter-propagating edge channels on R_{xx} . In the situation shown in the right inset of Fig. 4b, the antidot has become so large that the counter-propagating edge channels are fully reflected. In a naïve picture, this is equivalent to having no counter-propagating edge channels, meaning that R_{xx} should be zero. This is the case when all the edge channels, including the upstream one shown in blue, are connected to ohmic contacts. In contrast, in the experiment shown in Fig. 4b, R_{xx} is finite when the counter-propagating edge channels are fully reflected. A similar behavior is reproduced in a tight-binding simulation. However, the correspondence between the geometry used for the simulation (Fig. 4d and Fig. S8c) and the actual experimental configuration (Fig. 1a) is unclear. Although supplementary Fig. S8c explains this point, it is not clear enough. As I understand, the picture of quantum Hall effects in Ref. 22 requires that the upstream edge channel is decoupled from ohmic contacts. Then I wonder how one can simulate transport in such a situation. Perhaps, adding ohmic contacts, voltage source, and the ground to Fig. 4d and Fig. S8c might help. (Also, the ohmic contacts in Fig. 4d should be indicated by color distinct from red and blue.)

As pointed by Reviewer #1, the schematic drawing presented in Fig. 4b is too simplistic. We indeed neglected the fact that, when a QHEC is fully reflected, then, R_{xx} falls to zero. We have therefore corrected this drawing. Furthermore, we have added new figures and text in the supplementary materials (section S7 in the new version), where we develop the different regimes of coupling between the down- and upstream QHECs.

The remarks exposed by reviewer #1 concerning the crucial role of the ohmic contacts and their (de)coupling with upstream modes is also fully relevant. As indicated in the answer to comment (1), we have developed this aspect in more detail in a second article [arXiv:2103.10331]. In the latter paper, we come to the same conclusion as Reviewer #1: the upstream QHEC is not well coupled to the ohmic contacts. However, to reach that conclusion, we need to take into account a more complex picture involving local doping in the vicinity of contacts and different scenarii of backscattering along the sample borders, as well as simulations involving the whole device, and not just a small part of it (as in Fig. 4d). We therefore prefer to leave this more complex discussion in this new preprint.

Nevertheless, we believe the simulations performed in the present article are still fully valid, considering the physics at play. Indeed, they give a good insight in the mechanisms coupling the up- and downstream QHECs through the antidot. More precisely, simulations give a proper description of the transmission $T = T_{da} \times T_a \times T_{au}$, where T_{da} and T_{au} are the transmission arising from the overlap of the antidot QHEC with the down- and upstream QHEC respectively and T_a is the transmission through the antidot (see Fig. S7 of the supplementary). In particular, $R_{xx} = 0$ when T_{da} , T_a or $T_{au} = 0$ or if $T_{da} = T_{au} = 1$. In the other cases, R_{xx} is different from zero. This phenomenology can be reproduced in simulations assuming that ohmic contacts for current injection are ideally coupled with up and downstream modes, ohmic contacts for R_{xx} measurements are coupled to downstream modes (and to upstream modes when antidot backscattering comes into play), and by considering only a small region of the sample around the antidot. Note that the R_{xx} probes are now located on the opposite side of the sample in the new version of Fig. 4d. The simulated R_{xx} is exactly the same as in the former version, but this may help lift some confusion in the reader's mind.

We have also added some text in the supplementary materials to clarify the correspondence between the simulation results and the experimental configuration.

Note that the color of contacts has been modified in Fig. 4d.

3) The scanning gate images reported in this manuscript show only features related to antidots. However, in principle, just a tip-induced potential can also affect the scattering between counter-propagating edge channels, which should emerge in scanning gate images as in Ref. 23. Why are only features due to antidots visible? Is it because the authors are focusing on the filling factor region where the incompressible strip is very wide? The authors should elaborate on this point, together with the connection to other reported experiments.

To answer this question, we must consider three different aspects:

- In Ref. 23, the tip potential is used to induce a direct coupling between up- and downstream QHECs. Due to the low magnetic field used in the latter experiment (around 1 T), the tip-induced potential perturbation is much higher than the energy spacing between the Landau levels in their study than in our case, where we use a magnetic field above 8 T (up to 14T). Furthermore, the QHECs at $B = 1$ T are wider (since their width depends on the magnetic length l_B) and therefore overlap (see section S2 of the supplementary in [arXiv:2103.10331]). In turn, this yields a measurable coupling without the tip ($R_{xx} > 0$), while in our SGM data, $R_{xx} \sim 0$ when the tip is far away from antidots), so the presence of the tip potential (even small) simply tunes the coupling. The result is the appearance of iso-resistance stripes along the device edges. In contrast, the signature that we have observed (namely concentric rings of higher R_{xx}) confirm that the tip slightly changes the charge state of the preexisting antidot and possibly couple the up- and downstream QHECs. We have added some sentences in page 3 of the manuscript to clarify this important difference: “[...] *These antidots are at the origin of the characteristic concentric rings of non-zero R_{xx} in Figs. 1c-e. Note that these SGM signatures do not originate from a direct coupling of the counterpropagating QHECs induced by the tip potential alone: this would yield iso-resistance stripes following the edge topography [Marguerite Nature (2019), Tomimatsu PRR (2020)]. The absence of such stripes in SGM maps (Figs. 1) testifies that the tip perturbation is small enough to avoid inducing direct backscattering.*”
- The width of the tip perturbation (from Fig. 4c: around 400 nm) is much larger than the antidots and is also larger than the estimated distance between the up- and downstream QHECs. By applying stronger tip values, we therefore *shift* all the potential landscape below the tip with respect to the Fermi energy since the tip-induced potential simply adds to the antidot potential. A strongly biased tip will therefore induce the coupling between up- and downstream QHECs more easily at the antidot location. This is highlighted by Figs. 1c and 4a, where a large V_{tip} is shown to induce the coupling between the up- and downstream QHECs (indicated by high R_{xx} spots) more efficiently in the vicinity of the antidots. Obviously, this is also why, in [Marguerite Nature (2019)], they observe circular features centered along the device's edges in their SGM maps.
- We indeed focus on filling factor region where the incompressible strip is wide, since most of our data were obtained at the onset of a plateau ($R_{xx} \sim 0$). However, Fig. S3 shows SGM maps for different widths of the incompressible strip (by moving off the plateau) and it appears that antidots remain the main origin of coupling between the up- and downstream QHECs (*except on Fig. S3f, measured on the maximum of resistance on Fig. S3a: in this case the contrast is mainly observed at the center of the constriction*). In particular, we do not observe any uniform increase of R_{xx} when the tip is close to the edges. The increase of R_{xx} is associated with the signatures of antidots no matter the considered V_{bg} .

4) (optional) The authors emphasize that contrast in scanning gate image appears at a large distance

from the constriction to highlight their findings. However, it appears to me that the explanation starting from constriction makes it more difficult instead to understand what the authors are actually looking at. Firstly, this is because the mechanism for the R_{xx} increase is different for the backscattering at the constriction and that between counter-propagating edge channels on one side. Secondly, from Fig. 1a the readers would expect that something is happening at the constriction, but actually, it does not play a significant role, as the authors state. (It is not clear from reading whether the constriction plays any role at all, though. This should be clarified. Perhaps it does not play any role here.) Thirdly, the possibility of non-topological counter-propagating edge channels is already mentioned in the introduction. So, the authors do not need to start from “the textbook framework of QH effect.”

We understand the referee’s concern, but on the other side, we think that “hiding” the presence of the constriction at the beginning of the paper would also appear as suspect in the reader’s eye (at some point we would anyway have to show it, e.g. in Fig. S3). We believe that the observation of contrast away from a constriction is still surprising at this stage (considering that the paper of Marguerite et al. [Marguerite, A. et al. Nature 575, 628 (2019)] is still relatively recent, and no other papers report on such data). Furthermore it is also important as a way to clearly emphasize that the situation is different from the one in semiconductor-based 2DEGs. Constrictions are also usual building blocks of quantum Hall interferometers (Fabry-Pérot, Mach-Zehnder), so we prefer to also evidence that their role may also be completely different in the case of graphene QH systems.

5) It is hard to tell what the blue and purple circles above Figs. 3e, f represent. I guess these correspond to range of V_{bg} shown by the blue and purple arrows in Fig. 3c. Since these colors are similar to the blue and red colors representing the upstream and downstream edge channels, alternative colors should be used.

We have changed the colors in Figs. 3e,f to green, in order to avoid the risk of confusion pointed out by Reviewer #1.

Reviewer #2 (Remarks to the Author):

The authors employ a scanning gate microscope to study the graphene transport in the quantum Hall regime. They find that the QH transport can be perturbed by the tip, and they explain the behavior by considering an antidot near the graphene edge that could mediate backscatterings between counter-propagating QH edge channels formed due to electrostatic potential. While I appreciate the interesting data on probing such impurities, I have a few concerns about the general implications of these results. I am not convinced that this paper meets the broad audience criterion for publication in Nature Comm.

We thank Reviewer #2 for his positive comments about our data. We hope that our answers to his questions, as well as the modifications brought to the manuscript intended to emphasize the generality of our results will convince Reviewer #2. In particular, as detailed in the answer to the first comment of Reviewer #1, we added several sentences connecting our observations on antidots to STM observations of spatial inhomogeneities in Dirac point position in energy in graphene/hBN [J. Xue et al., Nature Mat. 10, 282 (2011)]. We also emphasized the role of fringing fields and impurities at etched edges of heterostructures, leading to the emergence of upstream modes, the second important ingredient in our model.

Except some particular cases (devices with graphite backgate, or defined by natural edges of flakes), our model based on antidots and upstream modes, and our conclusions apply to all van der Waals heterostructure-based devices, so we believe it should reach a wide audience.

1. The presence of such local potential perturbation in graphene, whether it is due to defects or

charge impurities, is not surprising. The question is how they would affect the QH transport. The paper's title says "Upstream modes and antidots poison graphene quantum Hall effect", but it seems that such poisoning occurs only when a tip is near the edge. What would be the effect of such antidot on QH transport without a tip present? I think this is an important question for the broad audience to know.

It is important to understand that the poisoning of QHE by antidots exists without the tip. SGM is used to locate the antidots and study how they couple the upstream modes. While this is true that SGM contrast is achieved by changing the charging state of antidots with the tip, it does not mean that antidots do not play any role without the tip. Varying the back gate voltage V_{bg} also change the charging energy of antidots and hence, the coupling between upstream modes. This is clearly demonstrated in data presented in Fig. 3 and in the associated discussion: peaks emphasized by the colored dashed lines in the R_{xx} vs V_{bg} curve in Fig. 3d are visible when the tip is far away from the antidots, and moving the tip just reveal their origin. Here is an extract from the manuscript summarizing the role of the tip: *"Importantly, the Coulomb resonances are also observed when the tip is far away from the device edges which means that the tunneling through the antidot is not necessarily triggered by the tip potential. Indeed, the Coulomb resonance signatures can be tuned by V_{bg} as shown in Fig. 3d."*

To clarify the role of tip, we have added some sentences on these aspects in page 3 of the manuscript:

"[...] These antidots are at the origin of the characteristic concentric rings of non-zero R_{xx} in Figs. 1c-e. Note that these SGM signatures do not originate from a direct coupling of the counterpropagating QHECs induced by the tip potential alone: this would yield iso-resistance stripes following the edge topography [Marguerite Nature (2019), Tomimatsu PRR (2020)]. The absence of such stripes in SGM maps (Figs. 1) testifies that the tip perturbation is small enough to avoid inducing direct backscattering."

2. In the schematics in Fig.2, a current loop is drawn around the antidot. I am a little confused about this. Does the loop represent the QH edge channels in the antidot? If so, the scenario in Fig. 2b should also cause backscatterings, through the edge channels around the loop.

The artist' view of Fig. 2 is obviously simplified. The loop represents one of the available discrete energy levels of the antidot, which takes the form of a circular QH edge channel, and can only contribute to backscattering if its energy is aligned with the Fermi energy. In the case of Fig. 2b, there is no discrete level aligned with the Fermi energy : the red dashed line is not aligned with any horizontal dark lines, corresponding to the available "discrete QH edge channels levels" pinned by the antidot potential. More theoretical details can be found in refs [25-27,30-31] of the manuscript about the effect of spatial confinement induced by the antidot potential in a quantum Hall system.

3. It is known that in the QH regime, electrostatics can cause reconstruction and form incompressible regions (PRB 46, 4026 (1992)). What is the role of such incompressible region in the present case?

The question of incompressible regions addressed in PRB 46, 4026 (1992) has improved the understanding of QHE by proposing a more accurate picture than the naive one of 1D edge channels widely used, still today. This picture has also been theoretically investigated in graphene, in presence of edge reconstruction (PRB 77, 155436 (2008)). Due to the sharp confinement potential at the device edges, the compressible and incompressible strips associated to downstream QH channels are much thinner than the magnetic length. Hence, the 1D picture of QH edge channels is perfectly valid for downstream modes.

The increase of potential along the edges associated to the screening of the back gate potential is however smoother. The resulting upstream modes would then be better described by a compressible strip separated from the downstream modes by an incompressible region, as represented in Fig. 3g. The QHECs loops pinned at antidots would also be better represented by circular compressible

strips. We however chose not to enter into these details and kept the 1D picture for the following reasons:

- What matters is the width of the incompressible region (distance between the QH edge channels in the 1D picture). Indeed, it determines the coupling strength between the QH channels and the antidots. Since we can not extract precise quantitative data on the antidots size and on the exact potential shape at the edges, one can not really distinguish between the two pictures (1D QH channels or compressible strips).
- In the simulations, the wavefunctions associated to the 1D QH channels have a given spatial extension of the order of the magnetic length. Considering compressible strips would simply increase the width of the wavefunctions. Since the coupling between QH channels is given by the overlap of these wavefunctions, the important data is the distance between the QH channels, and we do not have any quantitative information on this point, as written above.
- The 1D QH channels picture remains widely used and allows an easier visualization of the physical system.

Nevertheless, we qualitatively explored how compressible strips would influence the results of our simulations. To do so, we modified the potential at the edges as shown in Fig. C2. The result are similar with or without compressible strips except that the peaks are more pronounced with compressible strips since the distance between QH channels (incompressible regions) are smaller and then, the coupling larger.

Figure C2 : Effect of compressible strips in simulations. **a**, Potential $U(r)$ that mimics the enlargement of QH channels into compressible strips due to screening effects. **b**, The solid lines represent the LLs which follow the potential profile obtained along the dotted line **a**. The LLs are flattened where the $n = -1$ crosses the Fermi energy E_F . The dashed lines represent the LLs without incompressible strips. **c**, R_{xx} as a function of U_{bulk} (see Fig. S8d) obtained with compressible strips (e.g. potential in **a**). The red dashed lines indicate the theoretical positions of R_{xx} local maxima (alignment of E_F with $n = -1$ and $n = -2$ LLs) without bending of the edge potential. **d**, Solid curve : R_{xx} as a function of the maximal tip-induced change of hole density Δn_{tip} obtained with compressible strips (solid line in **b**) for $U_{bulk} = 0.102$ eV (red arrow in **c**). Dotted curve: same as solid line but without compressible strips (dashed line in **b**). The green arrow highlights the position of the first R_{xx} peak.

The tip voltage can induce a local charge accumulation and create incompressible rings around such charge bubble in the QH regime, as demonstrated by a previous SPM experiment (PRL 95, 136804 (2005)). The authors should consider if and how this effect affects the analysis.

The discussion presented in the PRL 95, 136804 (2005) does not affect our conclusions for three main reasons:

- The characterization technique is really different from SGM. In their study, they indirectly measure the conductance between the bulk region of a 2DES and the center of the charge bubble through the incompressible ring. To do so, the capacitance between the tip and the device is measured.
- In the PRL, the device's bulk is conductive and the tip-induced potential perturbation amplitude is close to the energy spacing between two Landau levels, in order to create the

charge bubble surrounded by the incompressible ring. In our case, the tip is used as a small perturbation to the potential, i.e. smaller than the energy spacing between two Landau levels. Hence, we do not induce the same kind of charge bubble, surrounded by an incompressible ring, and the physics discussed in the PRL paper is not relevant in our case.

- Here, the tip does not create a charge bubble but only modifies the width of an existing antidot or the distance between up- and downstream QH edge channels. In other words, it acts in a similar way as the back gate potential but over an area of ~ 500 nm centered below the tip. This is why the signatures associated to antidots are visible without the tip (or with the tip far away from antidots), as explained in the answer to question 1.

5. I think Fig. 3e and 3f are misleading. These two are supposed to describe the scenario around $\nu=-6$ plateau. In Fig. 3e, the bulk E_f is between $n=0$ and $n=-1$, which should correspond to $\nu=-2$ plateau. In the QH regime, the chemical potential is not linear in terms of density or gate. Also, what are the tilted lines in Fig. 3c at $V_{bg} < -21.5V$? Why do these features depend on tip position?

Considering that the upstream QHEC is not perfectly coupled to the contacts, the filling factor is not given by the bulk but by the incompressible region between the up- and downstream QHECs. We have added a paragraph in the manuscript to explain this and we have recently posted the preprint of an article focusing on the coupling between the upstream QHEC and the contacts [arXiv:2103.10331], which discusses these aspects and clarifies the discussion.

Concerning the tilted lines in Fig. 3c, at $V_{bg} < -21.5V$, they correspond to signatures of the topological breakdown due to the appearance of a third downstream mode and its corresponding upstream mode (between $\nu=-6$ and $\nu=-10$ plateaus). This breakdown occurs away from the antidot we are studying. However, even when the tip is far away, it has an influence on the scattering center at the origin of the topological breakdown. Hence, the V_{bg} values at which occurs the breakdown are slightly shifted when changing the tip position. We have added a discussion on those signatures at the end of section S2 of the supplementary materials.

6. In Fig.4a, what sets the polarity of the observed parabola features? I suppose similar parabola facing downward can be observed on the negative tip voltage side but perhaps at a different back gate? The work function difference of 3V between tip and graphene seems to be rather large. What is the material of the metal on the tip?

The polarity of the parabola is determined by the tip voltage. Parabola going downwards are expected for strongly negative tip voltage, as observed for example in another study on quantum dots at zero magnetic field (PRL **93**, 216801 (2004)), at any back gate voltage. The work function difference of 3V is indeed high (it is about 0,5V for sample G2) since the tip is a platinum-iridium one. However, the device is found on the sample by scanning metallic marks in topographical mode with the tip, and it can cause contamination of the tip with other metals, as gold is used for metallic marks, and change the work function (gold is known to have a large workfunction). The tip was however imaged with an electron microscope after the experiment and was still very sharp, suggesting that it was not severely damaged along the experiment, but small particles of gold may have stucked to the tip.

Reviewer #3 (Remarks to the Author):

The goal of the paper "Upstream modes and antidots poison graphene quantum Hall effect" is the experimental investigation of the instability of Quantum Hall edge transport in realistic graphene samples. Authors defend the scenario there electrostatic accumulation of charges at the edges of graphene flake results in existence of both downstream and upstream counter-propagating edge channels in the integer Quantum Hall effect regime. Then authors suggest that special impurities "antidots" which happen to exist near the edges of graphene would lead to electron's scattering into the upstream channels, breaking effectively the topological protection of the current.

Using the scanning gate microscope authors were able to identify such antidots and to measure directly the current distribution inside the impurity/antidot and in the both downstream and upstream channels. To my opinion this is an interesting result, which deserves to be published.

Below I give a list of questions, which I suggest to consider before publication.

We thank Reviewer #3 for his positive feedback. We hope that the answers to his questions address all his concerns.

1. Authors say (page 3 right bottom) that the distance between antidots is 100-500 nm. Where this comes from? Is it the bulk distance between randomly distributed antidots, or the typical distance between the impurities along the edge?

This distance corresponds to the typical distance between observed antidots along the edges, visible in our SGM images. The distance between impurities can not be known from our data since only the active antidots are spotted (in Figs S3 of the supplementary materials, more and more antidots are activated by increasing V_{bg} for instance). Nevertheless, our observations are consistent with the typical lengthscale of potential fluctuations observed in graphene on hBN through STM characterization. We have added several sentences about it in the manuscript (see the answer to the first comment of Reviewer #1).

2. Am I right that all the experiments were performed in the hole doped graphene (negative bias in Fig. 1b)? Why? And may authors say that explicitly?

All the measurements presented in the paper, for the two studied samples, were indeed performed in the hole side. For the two samples, this is due to technical issues:

- For the second sample (data presented in the supplementary materials), the charge neutrality point was shifted to +7V and the backgate was leaking around +11V. It was therefore not possible to explore the electrons side at high magnetic field.
- For the first sample, the contacts were not working properly on the electrons side above 4T, as can be seen in Fig. S1a of the supplementary materials. For these parameters, the longitudinal resistance exhibits large fluctuations due to bad equilibration between the quantum Hall edge channels and the contacts and the current measured through the sample fell down to zero at low electrons density. It prevented us to obtain clean SGM maps at high magnetic field, equivalent to those obtained for holes. However, we made some measurements on the electrons side at lower magnetic field (around 4T). This is the topic of another article [arXiv:2103.10331] where we discuss in particular the asymmetry between electron and hole side.

3. Why the impurities are called antidots? In usual semiconductors with parabolic dispersion one indeed needs antidots for similar effects. But in graphene with its Dirac spectrum, what is a quantum dot for electrons became an antidot for holes.

An antidot for holes is indeed a dot for electrons. Here we have chosen to use the term "antidot" defined as a local decrease in holes density (which gives possibly rise to a quantum Hall channel loop at high magnetic field), and for electrons, as a local decrease in electrons density. The rationale behind the choice of the term "antidot" is essentially to make a bridge with the literature produced for semiconductors-based heterostructures, since the physics is similar.

4. I think, there is a related physical question. Authors confess, that the exact nature of impurities-antidots is unknown. One possibility is that they are due to the charged impurities in hBN layer. In that case they may have the charge of always the same sign. Is that so? From the first experiments in graphene the Quantum Hall effect was observed for both Fermi energy below and above the Dirac point. But the electrostatic charge accumulated at the edges would have opposite sign in these two cases, while the impurity potential would stay the same. Would that

make the effect particle-hole asymmetric? Can this be seen and used to support the picture suggested by the authors?

We also considered the possibility of charged impurities located in hBN. Unfortunately, we didn't find any paper in the literature that has studied extensively defects in hBN and their effect on graphene. We are therefore unable to answer rigorously the first question, regarding the charge sign of such impurities. However, one must notice that the Dirac point in the first sample is almost not shifted. It suggests that there is not a large amount of impurities (or that the amount of impurities of both polarities is equilibrated) in hBN.

Regarding the second part of the question, the idea is really good. By performing such measurements in the electrons side, an asymmetry in the results could help figuring out the kind of impurities at the origin of antidots. However, we managed to demonstrate in [arXiv:2103.10331] that such asymmetry likely originates from the contacts and in this framework we could not extract information about the type of impurities.

REVIEWERS' COMMENTS

Reviewer #1 (Remarks to the Author):

I am happy with the revised manuscript, where the authors have appropriately addressed the issues I raised in my previous report. I recommend it for publication.

Reviewer #2 (Remarks to the Author):

My questions are addressed. I recommend publication.

Reviewer #3 (Remarks to the Author):

I have read carefully the answers given by the authors to my and other referees comments. I am satisfied with this response and believe, the paper may be published now.

I have only two small comments/requests. I am satisfied with how authors answer to me. But I expected, that they would also clarify the issues touched by me to the readers also.

Specifically, my question 2. Can authors say just few words explicitly in the text that their two samples were functioning only on the n-doped side? Or there are such words somewhere, which I missed?

Similarly, my question 3. May authors say explicitly that their antidots are antidots only for holes (would become quantum dots for electrons).

I believe, both suggestions may cost just several words in the minimal version.

Reviewer #1 (Remarks to the Author):

I am happy with the revised manuscript, where the authors have appropriately addressed the issues I raised in my previous report. I recommend it for publication.

Reviewer #2 (Remarks to the Author):

My questions are addressed. I recommend publication.

Reviewer #3 (Remarks to the Author):

I have read carefully the answers given by the authors to my and other referees comments. I am satisfied with this response and believe, the paper may be published now.

I have only two small comments/requests. I am satisfied with how authors answer to me. But I expected, that they would also clarify the issues touched by me to the readers also.

Specifically, my question 2. Can authors say just few words explicitly in the text that their two samples were functioning only on the n-doped side? Or there are such words somewhere, which I missed?

We have addressed this point in the following sentence, in the new version of the manuscript, and we refer to the supplementary materials where this is further developed:

*"Here, we use scanning gate microscopy (SGM) to build a full microscopic picture of QHECs topological protection breakdown in graphene. For this purpose, we studied two devices (G1 and G2), consisting in 250 nm-wide encapsulated graphene constrictions as presented in Fig. 1a **and functioning only in the p-doped side at high magnetic field (see supplementary section S1)**"*

Similarly, my question 3. May authors say explicitly that their antidots are antidots only for holes (would become quantum dots for electrons).

We have added a new sentence to address this issue:

*"More realistically, such potential landscape could originate from two known possible sources: (i) nanoscale random strain fluctuations, known to induce charge density inhomogeneities in graphene (ii) remote charged impurities in the dielectric hBN layer. Both sources lead to local variations of Dirac point energies (typically ~50-100 meV at B=0 T, over typical distances ~ 50-100 nm, probably ubiquitous in all hBN/graphene/hBN heterostructures. **It is noteworthy that a potential fluctuation giving rise to an antidot on the p-doped side would yield a dot on the n-doped side.**"*